# Sequelae following an epidemic of meningococcal meningitis in Niger in 2022

**Abdoul-Aziz Idrissa[1], Salifou Atti[2], Roger Kiamvu Wasaulua[3], Serge Kazadi[4], Ousmane Sani[5], Elhadji Ibrahim Tassiou[6], Ousmane Guindo[7], Georges Tonamou[8], Iza Ciglenecki[9], Matthew E. Coldiron[10]\***

1 Epicentre, Maradi, Niger, 2 Ministry of Public Health, Magaria, Niger, 3 Médecins Sans Frontières, Magaria, Niger, 4 Médecins Sans Frontières, Niamey, Niger, 5 Centre de Recherche Médicale et Sanitaire, Niamey, Niger, 6 Ministry of Public Health, Niamey, Niger, 7 Epicentre, Niamey, Niger, 8 Médecins Sans Frontières, Dakar, Senegal, 9 Médecins Sans Frontières, Geneva, Switzerland, 10 Epicentre, Paris, France

* matthew.coldiron@london.msf.org (MEC)

## Abstract

### Background

The management of post-meningitis sequelae is a priority in the WHO Roadmap to Defeat Meningitis by 2030. Nonetheless, the prevalence of sequelae in the African Meningitis Belt is not well described, making the development of post-meningitis care programmes difficult. We conducted a home-based follow-up study of cases notified during an epidemic due to *Neisseria meningitidis* serogroup C (NmC) in 2022 in the Dungass and Magaria Districts of Niger, to describe the prevalence of sequelae several months after the epidemic.

### Methods

Standard WHO case definitions were used during the epidemic. District linelists were completed with the results of PCR testing of patients who had undergone lumbar puncture. These lists included the village of origin of the notified cases. Accompanied by community outreach workers, the study's nurse-investigators sought out case-patients in their homes to assess the presence of sequelae. A standardised questionnaire was administered, and a focused physical examination was carried out.

### Results

A total of 1001 suspected cases and 50 deaths (CFR 5.0%) were reported in the two districts. A total of 469 CSF samples (47%) were analysed at the national reference laboratory, of which 220 were PCR positive (47%). NmC was the predominant causative organism (87% of confirmed cases). 82 cases were excluded due to distance. 570 of the 919 cases sought out were eventually found and included. Of these 570 cases, 49 had died (CFR 8.6%). Among surviving cases, the prevalence of sequelae

**Data availability statement:** All relevant data are within the manuscript and its Supporting Information files.

**Funding:** Funding for the survey was provided by Médecins Sans Frontières.

**Competing interests:** The authors have declared that no competing interests exist.

was 12%, and among survivors of confirmed NmC meningitis, 18%. The most common sequelae were hearing loss (6%), paralysis (3%) and epilepsy (2%).

## Discussion

Case fatality during the epidemic appears to have been higher than reported in routine surveillance. The prevalence of severe sequelae is high, and clinical description of sequelae could help future epidemic management.

## Introduction

The causative organism of meningococcal meningitis is the gram-negative diplococcus *Neisseria meningitidis,* which is transmitted between humans via respiratory droplets [1]. Several serogroups of *N. meningitidis* exist. Historically, *N. meningitidis* serogroup A (NmA) was the most important causative agent in Africa, but other serogroups such as NmX and NmW have caused epidemics [2–4]. In recent years, a new serogroup C strain has emerged, leading to major epidemics in Nigeria since 2013 and Niger since 2015 [5–7]. Meningococcal meningitis is characterised by a meningeal syndrome: typically presenting with sudden onset of fever with headache and stiff neck. There are systemic symptoms, but the prevalence of meningococcemia (often with disseminated intravascular coagulation and its dermatological manifestations) is less frequent in the Sahel than in western countries [8]. With appropriate antibiotic treatment, the case fatality rate associated with meningococcal meningitis is normally between 5–10% [1].

Classically, the rate of severe sequelae among surviving cases of meningitis in Africa was thought to be low [9]. Relatively few studies have been carried out in Africa, but more recent studies show that the burden of sequelae is significant. In Senegal outside of epidemics, half the children affected by all-cause meningitis had a hearing impairment, 40% had cognitive problems, and 20% had epileptic seizures after a case of meningitis [10]. That said, the study followed patients with any cause of meningitis - including pneumococcus and *Haemophilus influenzae*, two organisms associated with a higher rate of sequelae than meningococcus. In Niger, after the 2015 meningococcal meningitis epidemic in the district of Doutchi, sequelae were observed in 12% of patients, with the most common being impaired hearing and cognitive impairment [7]. The use of corticosteroids, especially dexamethasone, as adjuvant treatment to reduce sequelae remains controversial. In theory, corticosteroids could reduce the inflammatory response, perhaps leading to a reduction in the prevalence of sequelae. However, this approach is currently only recommended in Western countries, and only for pneumococcal and *H. influenzae* meningitis. The empiric use of corticosteroids is not recommended in Niger, and any use is at the discretion of the treating clinician. There are currently no guidelines for the screening, referral or treatment of meningitis sequelae in Niger.

A NmC epidemic, as defined by WHO thresholds of >10 cases/100 000 persons/week occurred in parts of the Zinder Region of Niger in 2022 [11]. The Magaria and

Dungass Health Districts (HD) were the most affected, with >1000 cases notified from 20th November 2021 (the start of the 2022 epidemic season). A mass vaccination campaign targeting people aged between 2 and 29 years of age with a polysaccharide ACW vaccine (vax-MEN-ACW®, Finlay) was carried out from 17th March 2022. No formal vaccine effectiveness evaluations were carried out in the study area following these campaigns.

We performed a descriptive study to complete the clinical and microbiological description of the meningitis epidemic in Magaria and Dungass Districts in 2022. The study consisted of a follow-up visit to case-patients in their homes several months after their acute episode of meningitis. This study would permit an estimation of the prevalence of sequelae, as well as a description of the risk factors for the development of meningitis, particularly in households with a case.

## Materials and methods

### Case definition

The definitions of cases of meningitis used during the epidemic were as follows [11]:

- Suspect case (in children aged ≤1 year): Sudden onset of fever (≥38.5°C rectal temperature or ≥38.0°C axillary temperature) with at least one of the following signs: stiff or flaccid neck, bulging fontanel, flattening of the eyes, convulsion or any other meningeal sign.

- Suspect case (≥1 year old): Sudden fever (≥38.5°C rectal temperature or ≥38.0°C axillary temperature) with at least one of the following signs: stiff neck, neurological disorder or any other meningeal sign.

- Probable case: Any suspect case in whom the LP reveals CSF with a macroscopic appearance that is cloudy, purulent or xanthochromic, or the presence of Gram-negative diplococci, Gram-positive diplococci or Gram-positive bacilli on microscopic examination, or if the leucocyte count is greater than 10 cells/mm3.

- Confirmed case: Any suspect case in which the causal agent has been demonstrated by culture or PCR of CSF or blood.

### Study population

The study area corresponded to the Magaria and Dungass HD, and the study population corresponded to the population residing in these areas.

The inclusion criteria for the follow-up survey were as follows:

- Cases notified in health facilities in the study area according to the linelist of epidemic cases between 1st December 2021 and 30th June 2022

- Residing in the study area at the time of the survey

- Verbal informed consent given by the individuals themselves or their parents/guardians.

The only exclusion criterion was refusal to participate.

### Practical identification of cases

Reported cases were sorted by village of origin. The villages of residence of notified cases were mapped. However, for logistical reasons, the survey was not able to cover all cases in the entire Magaria and Dungass HDs. The study team therefore excluded villages that reported a single suspected case and that were more than 2 hours' drive from Magaria.

Members of the study team visited each selected village. On arrival, the investigators introduced themselves to the local authorities and asked for help in finding the homes of notified cases, with the help of community relays or other local guides. The survey was conducted between 16th and 30th November 2022.

### Biological data

No samples were taken as part of this study. The routine biological results provided by the national surveillance system were added to the information collected from participants who had received a lumbar puncture during the management of their disease.

### Data collection

Once inside the patient's home, the investigators asked to speak with the patient. After introducing themselves and explaining the objectives of the study, the investigators began the informed consent process. Once informed consent had been obtained, data was collected using tablet computers. A standardised questionnaire was administered by trained paramedical staff, by means of a face-to-face interview with the suspected case or with the person accompanying the case if the latter was unable to respond. The questionnaire contained demographic information (age, sex) about the case and their socio-economic status and place of residence, information about symptoms related to their meningitis episodes and the date of onset of signs, information about risk factors related to meningitis (contact with cases, smoking, travel, pilgrimage), the case's vaccination status and their health-seeking behaviours. The case's vaccination status for various meningitis vaccines was determined primarily on the basis of the vaccination card, or by self-report if no card was available.

### Special considerations for data collection

A significant proportion of the notified cases died during the epidemic, and others who survived suffered significant sequelae. A large proportion of the cases were minors, so measures were put in place to ensure that data was collected sensitively and effectively.

- Deceased patients - The investigators followed the same procedures described above, but introduced themselves to the head of the surviving household or his or her representative. After offering condolences, the interviewers offered the surviving person the opportunity to participate in the study. If the person did not wish to participate because of bereavement, this reason was noted and no further survey procedures were carried out.

- Patients with significant mental disability - Some case-patients with mental disability following meningitis were unable to provide informed consent or to complete the questionnaire. In these cases, consent was sought from the patient's guardian, who was also the respondent to the questionnaire.

- Minor patients - Standard procedures were followed, with consent provided by their parent or guardian, with assent for patients over 12 years of age.

   Patients who were lost to follow-up, or who could not be found by the investigation team after normal effort, were not further investigated.

### Data analysis

The demographic characteristics of the patients and their households, and the clinical characteristics of the patients were described using appropriate measures of centrality and dispersion. Stata 15.0 software (College Station, TX, USA) was used for statistical analysis.

### Ethical principles

The survey was conducted in accordance with the international ethical guidelines for biomedical research involving human subjects and the international ethical guidelines for epidemiological studies of the Council for International Organizations of Medical Sciences (CIOMS)[12]. The study protocol was approved by the *Comité National d'Éthique pour la Recherche*

*en Santé* (Ref: 40/2022/CNERS) and the MSF-ERB (Ref: 2260) before implementation. The survey was carried out with the agreement of the Niger Ministry of Public Health and the local health authorities in Magaria. Participants (or their guardians, or survivors of deceased cases, as described above) were asked to provide written informed consent prior to participating in the survey. The voluntary nature of their participation, confidentiality, and anonymity were emphasised to those responsible before they gave their consent.

## Results

### Epidemiological description according to official data

From 20th November 2021–30th October 2022, the Magaria and Dungass HDs reported 1001 suspected cases of meningitis and 50 deaths (CFR 5.0%), including 557 suspected cases and 37 deaths (CFR 6.6%) in Magaria and 444 suspected cases and 13 deaths (CFR 2.9%) in Dungass (Fig 1). At district level, the epidemic threshold was not crossed, although it was crossed in the Dungass and Bangaza Health Areas (HA) in the Dungass HD and the Dantchiao, Guetche and Dan Gouchi HAs in the Magaria HD (S1 Fig). In 4 of the 5 HAs that crossed the epidemic threshold, the vaccine response began >8 weeks after the epidemic threshold was first crossed. The cumulative attack rate was 67 cases per 100,000 in Magaria and 88 cases per 100,000 in Dungass. In the two districts as a whole, these 1001 cases were reported in 230 villages.

During the same period, the national reference laboratory received a total of 470 samples of CSF from the two districts (a sampling rate of 47% among suspected cases). A total of 220 samples tested positive for a bacterium causing meningitis (a positivity rate of 47% among the samples collected). The predominant causative organism was NmC (192 cases), followed by *S. pneumoniae* (22 cases), *H. influenzae* serotype b (4 cases) and NmX (2 cases).

Tables 1 and 2 show that, in both districts, the largest absolute number of suspected and confirmed cases were among children aged 5–14 years, although the attack rates and case-fatality rates were higher among children aged <5 years.

### Description of cases traced during the follow-up survey

Data collection took place from 16th to 30th November 2022 in 148 villages of the Magaria and Dungass HDs. The nurse-investigators attempted to locate and investigate 919 of the 1001 cases notified during the epidemic (82 cases were reported from villages reporting a single case and that were >2 hours' drive from Magaria or Dungass). A total of 570 cases were contacted at home (62% of those targeted) and all agreed to take part in the survey. Among all suspected cases traced during the survey, 49 had died at the time of the survey, including 43 who died during the epidemic and 6 after the epidemic period (i.e., an overall case-fatality rate of 8.6%, and 7.5% during the epidemic). The other 349 targeted cases could not be located using the information available to the teams.

At the time of the meningitis episode, a sample of CSF was collected and analysed from 356 of the 570 cases (62%) found during the survey. A total of 165 samples tested positive for a bacterium causing meningitis (45% of samples taken). Among the confirmed cases surveyed, the predominant causative organism was NmC (149 cases, including 11 deaths, i.e., a case-fatality rate of 7.4%), followed by *S. pneumoniae* (10 cases).

At the time of the survey, 27 cases that had occurred during the epidemic but were not notified in the linelist were found in the households visited, including 8 cases who had died. Overall, 24 of the 27 cases were treated in a healthcare facility; the 3 who did not receive care died at home in November and December 2021. Adding these cases to the study population gives an overall case fatality rate of 9.5% (57 out of 597). Given that no additional clinical information was available on these 27 patients, they are not included in the clinical descriptions below.

As shown in Fig 2 and Table 3, the characteristics of the case-patients successfully traced and investigated were generally consistent with the overall cases reported during the epidemic in terms of time (cf. Figure 1), as well as the distribution by age (cf. Table 1).

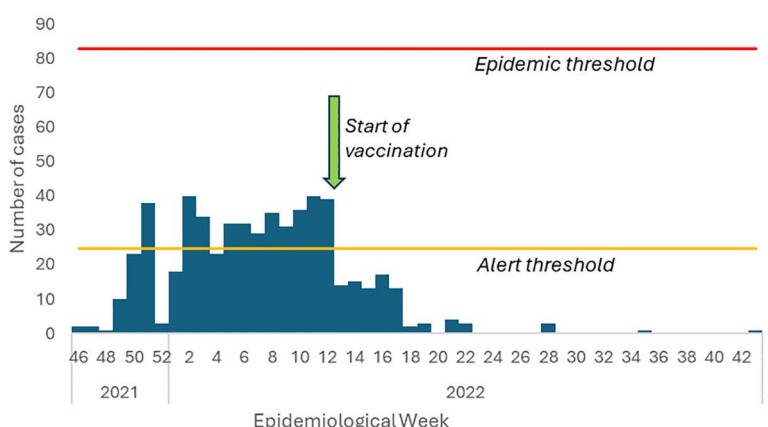

## Magaria Health District

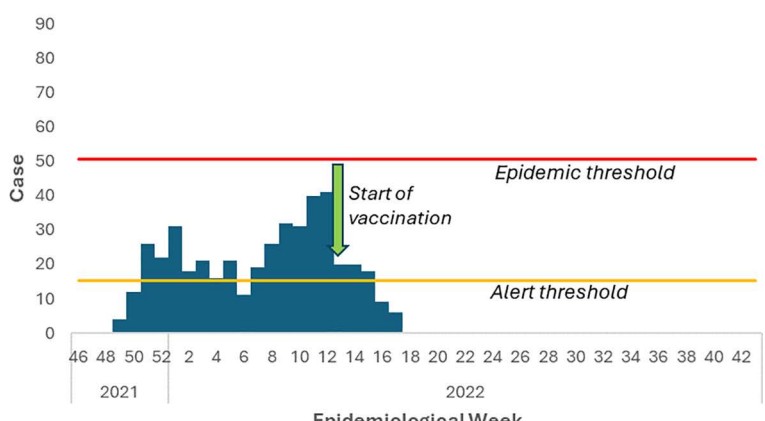

## Dungass Health District

**Fig 1. Suspected cases of meningitis, Magaria and Dungass Health Districts, 2021-2022.**

**Table 1. Distribution of meningitis cases by age, Magaria HD, Niger, 2021-2022.**

| Age (Years) | Suspected cases | | Confirmed cases | | Population | Attack rate per 100,000 | Case fatality (%) |
|---|---|---|---|---|---|---|---|
| | N | % | N | % | | | |
| 0-1 | 76 | 13.6 | 10 | 9.6 | 68240 | 111 | 5.2 |
| 2-4 | 129 | 23.2 | 15 | 14.4 | 93557 | 138 | 9.3 |
| 5-14 | 222 | 39.9 | 61 | 58.7 | 247257 | 90 | 7.6 |
| 15-29 | 86 | 15.4 | 15 | 14.4 | 220412 | 39 | 3.4 |
| 30-44 | 29 | 5.2 | 2 | 1.9 | 106526 | 27 | 3.4 |
| ≥45 | 15 | 2.7 | 1 | 1.0 | 92375 | 16 | 0 |
| All | 557 | | 104 | | 828367 | 67 | 6.6 |

**Table 2. Distribution of meningitis cases by age, Dungass HD, Niger, 2022.**

| Age (Years) | Suspected cases | | Confirmed cases | | Population | Attack rate per 100,000 | Case fatality (%) |
|---|---|---|---|---|---|---|---|
| | N | % | N | % | | | |
| 0-1 | 44 | 9.9 | 9 | 7.8 | 41797 | 105 | 4.5 |
| 2-4 | 114 | 25.7 | 22 | 19.0 | 57303 | 199 | 1.7 |
| 5-14 | 228 | 51.4 | 72 | 62.1 | 151445 | 151 | 3.1 |
| 15-29 | 49 | 11.0 | 12 | 10.3 | 135004 | 36 | 4.1 |
| 30-44 | 7 | 1.6 | 1 | 0.9 | 65248 | 11 | 0 |
| ≥45 | 2 | 0.5 | 0 | 0.0 | 56577 | 4 | 0 |
| All | 444 | | 116 | | 507374 | 88 | 2.9 |

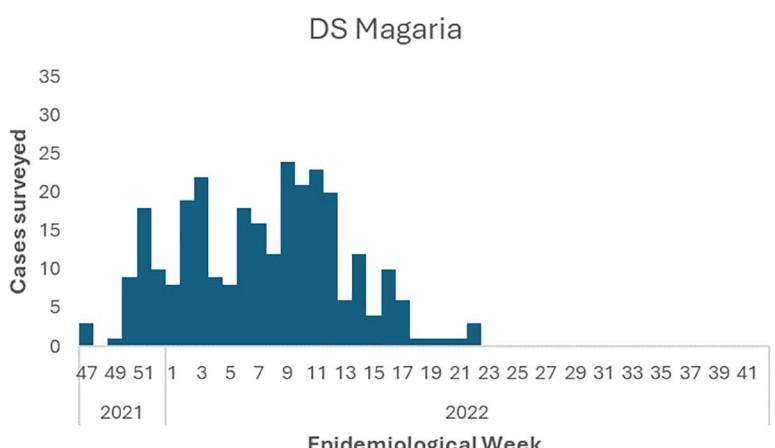

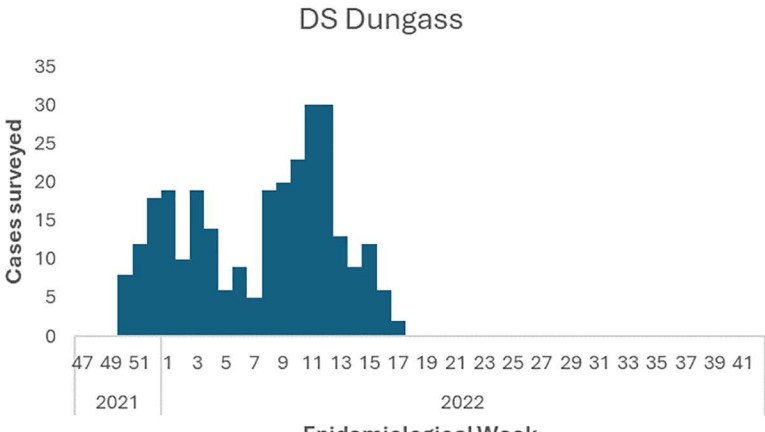

**Fig 2. Date of notification of cases participating in the follow-up survey, Dungass and Magaria Health Districts, 2021-2022.**

**Table 3. Age and sex of case-patients participating in follow-up survey, Dungass and Magaria Health Districts, Niger, 2021-2022.**

| Features | Cases surveyed (N=570) | | Confirmed cases surveyed (N=165) | |
|---|---|---|---|---|
| | N | % | N | % |
| Gender | | | | |
| M | 295 | 51,8 | 76 | 46,1 |
| F | 275 | 48,2 | 89 | 53,9 |
| Age (Years) | | | | |
| 0-1 | 67 | 11,8 | 17 | 10,3 |
| 2-4 | 128 | 22,5 | 32 | 19,4 |
| 5-14 | 275 | 48,3 | 90 | 54,6 |
| 15-29 | 72 | 12,6 | 18 | 10,9 |
| 30-44 | 18 | 3,2 | 6 | 3,6 |
| ≥45 | 10 | 1,8 | 2 | 1,2 |

### Risk factors for meningitis

Across the two districts, among all cases surveyed, 268 (47%) reported receiving no dose of meningococcal vaccine during the 2021 and 2022 campaigns, 152 (27%) one dose and 150 (26%) two doses. Among the 165 confirmed cases surveyed, 75 (46%) reported receiving no doses, 47 (28%) one dose, and 43 (26%) two doses.

Among the 32 confirmed cases aged 2–4 years, 21 (65%) reported having received at least one dose of vaccine in the 2021 and 2022 campaigns, and among the 90 confirmed cases aged 5–14 years, 54 (60%) reported having received at least one dose of vaccine in the 2021 and 2022 campaigns. Specific dates of vaccination were only available for 4 cases, all of whom received the vaccine after disease onset in 2022.

It should be noted that 66 of the 570 cases (11.6%) occurred in a household with >1 case notified during the epidemic, including 26 households with 2 cases, 3 households with 3 cases, and 1 household with 5 cases; in total, therefore, 534 households had cases. In these households with multiple cases, after notification of the first case, the attack rate among other household members was 1014 per 100,000 (36 additional cases among 3548 individuals at risk), i.e., >10 times higher than the attack rate among the general population of the two districts.

Within the 534 individual households reporting cases, contacts were asked whether they had taken antibiotic prophylaxis following notification of the case in their household. Prophylaxis was reported by at least one person in 215 of the 534 households (40%). In 171 of these households (80%), only one person received prophylaxis, so overall only 382 of the 4118 people (9%) living in a household with a case of meningitis took any prophylaxis. All of these contacts reported taking ciprofloxacin, 285 of the 382 as a single dose, and 72 for between 2 and 6 days (the duration of prophylaxis was not recorded for 24 people). There were no reported cases of meningitis among people who took ciprofloxacin. The study questionnaire did not ask about the source of the ciprofloxacin taken.

### Sequelae

Of the 521 cases notified alive at the time of the survey, 61 case-patients (12%) reported at least one sequela. The prevalence of sequelae was higher among confirmed cases of NmC (25/138, 18%, Table 4).

The most common sequelae were hearing loss and paralysis (Table 5). Hearing problems were reported in surviving cases of all ages (median age 10 years, IQR 4–15). Paralysis included various syndromes: 7 cases with paralysis of one or both legs, 3 cases of paraplegia, 2 cases with paralysis of one or both arms only, 2 cases of hemiplegia, and 1 case for which the syndrome was not recorded. The cases of epilepsy that developed after the meningitis were all reported in children, with 7 of the 9 cases being under the age of 10. In describing the cases of intellectual disabilities and regression,

**Table 4. Presence of sequelae in surviving cases of meningitis by age and causative organism, Magaria and Dungass, Niger, 2022.**

|  | Number of surviving case-patients | Sequelae present, n | Sequelae present, % |
|---|---|---|---|
| **Age in years** |  |  |  |
| 0-1 | 59 | 7 | 12 |
| 2-4 | 117 | 16 | 14 |
| 5-14 | 257 | 27 | 11 |
| 15-29 | 64 | 9 | 14 |
| 30-44 | 17 | 2 | 12 |
| ≥45 | 7 | 0 | 0 |
| **Laboratory confirmation** |  |  |  |
| NmC | 149 | 27 | 18 |
| *S pneumoniae* | 10 | 0 | 0 |
| NmX | 3 | 0 | 0 |
| Hib | 3 | 0 | 0 |
| PCR negative | 189 | 19 | 10 |
| PCR not performed | 216 | 21 | 10 |

**Table 5. Prevalence of specific sequelae among meningitis survivors in Magaria and Dungass, Niger, 2022. Numbers and % are not cumulative, as some patients had more than one sequela at follow-up.**

|  | Number of cases among all surviving case-patients (N = 521) | Prevalence among all surviving case-patients (%) | Number of cases among surviving case-patients with confirmed NmC (N = 140) | Prevalence among surviving case-patients with confirmed NmC (%) |
|---|---|---|---|---|
| Hearing loss/deafness | 29 | 5.6 | 14 | 10.0 |
| Paralysis | 15 | 2.9 | 9 | 6.4 |
| Epilepsy | 9 | 1.7 | 1 | 0.7 |
| Intellectual disability or regression | 6 | 1.2 | 3 | 2.1 |
| Severe headaches | 4 | 0.8 | 1 | 0.7 |
| Skin problems | 4 | 0.8 | 2 | 1.4 |

the respondents (in all 6 cases, a family member) described delirium, agitation, non-epileptic seizures, and in one case, simply that the child had "gone crazy".

## Discussion

### Overall aspects of the epidemic

An epidemic of NmC occurred in the Magaria and Dungass HD during the meningitis season in 2022, with >1000 cases notified in all districts. At district level, the epidemic threshold (10 cases/100,000/week) was not crossed. On the other hand, several HAs did cross these thresholds, underlining the importance of conducting meningitis surveillance on populations of <100,000 people, which makes the detection of local epidemics more sensitive [13,14].

The level of case confirmation during this outbreak was acceptable from an epidemic management point of view but fell short of the objectives of a case-by-case surveillance system, where almost all cases are expected to benefit from lumbar puncture and CSF examination. Almost half of the suspected cases underwent lumbar puncture, enabling NmC to be identified as the predominant causative organism and a reactive vaccination campaign to be organised with an

appropriate vaccine. Among the cases that were sampled, the positivity rate was around 50%, which is in line with historical data from epidemics in the African Meningitis Belt [15,16]. The aetiology of symptoms in these "negative" cases remains unknown and is probably due to several factors. Firstly, there may be a "broad" application of case definitions by health staff in peripheral facilities (which may be justified, especially in areas with limited access to care). Delays in sample storage and transport between the peripheral level and the reference laboratory may also have an impact on the quality of the CSF sample. Similarly, the PCRs used routinely only detect the most common bacteria - other common causative agents of meningitis, such as *Salmonella* and viruses, are not normally detected. In the absence of cytology and biochemical measurements, it is difficult to classify these "negative" cases.

Once again, during a meningococcal meningitis epidemic, at local level, the organisation of timely reactive vaccination campaigns proved complicated [17]. The rapid spread of epidemics, and their highly localised nature, mean that the epidemic peak is often passed before the vaccination campaign in the most affected health areas. With the introduction of a pentavalent meningococcal conjugate vaccine (with long duration of protection and significant indirect protection offered to non-vaccinated individuals) into routine vaccination programmes and for outbreak response, polysaccharide vaccines will likely have less use in reactive vaccination in coming years. During this epidemic, health staff provided some members of certain households with ciprofloxacin prophylaxis. This practice is not formally recommended during epidemics in Africa [18], and the sporadic way in which it is administered in this context limits the possibility of formally evaluating its real-world effectiveness. This is a missed opportunity. On the other hand, the fact that the attack rate among household members with a previously notified case was > 10 times higher than the attack rate among the general population highlights the value of providing household contacts of cases with protective measures, such as appropriate antibiotic prophylaxis (single dose of ciprofloxacin, shown to be highly effective at an individual level in areas with no meningococcal resistance to fluoroquinolones [19]) or early vaccination [20].

## Meningitis sequelae

This survey of cases, conducted several months after the epidemic, showed the high burden of sequelae among survivors, particularly hearing loss, paralysis and epilepsy. These sequelae are serious and often limiting. The prevalence of major sequelae described here is nevertheless lower than that described in the most detailed study in Africa, in Dakar (with a population of 66 cases - much fewer than our study), where the rate of sequelae was 65% in cases [10]. Several important differences between the studies should be noted: in the Dakar study, the study population was exclusively paediatric, and the cases were hospitalised in a referral hospital outside the epidemic period, and all the cases were confirmed bacterial meningitis, with the main causative organisms being pneumococci and Hib, with only 11 cases of *N. meningitidis*. In addition, in Senegal, researchers used formal scales to assess the presence of sequelae, whereas the methods used in the present study were more informal, and therefore possibly less sensitive, especially for minor sequelae. On the other hand, the prevalence of sequelae observed here aligns well with what was seen after the NmC epidemic in Doutchi, Niger, in 2017, when the prevalence of sequelae among all cases was 11%, and 16% among confirmed cases of NmC [7]. Nonetheless, we note that the rate of deafness was 10% and paralysis 6.4% among surviving patients with confirmed NmC meningitis. Although the methods used in Doutchi and 2017 and in Magaria in 2022 are very similar, which could probably have influenced the results, the differences seen with the study in Senegal are probably more related to the different contexts and different epidemiological factors.

The methodology used in the follow-up survey merits discussion. The investigators were only able to find 62% of the targeted cases, and this after having discarded 8% of the notified cases for logistical reasons. In the end, the 570 cases surveyed represented only 57% of the cases notified during the epidemic. First of all, the fact that the investigators were able to find so many cases - when they started searching for cases 9–11 months after the epidemic, when they had only

surname, first name and village of origin - exceeded pre-survey expectations. By all measures, the cases surveyed seem similar to all notified cases, at least in terms of the factors that could be measured: in terms of age and sex, case confirmation and causative organism, and in terms of the date of case notification. The fact that case fatality was higher among surveyed cases reflects inaccuracies in the surveillance system – which is not designed for repeated, longitudinal data capture among patients hospitalized for several days or weeks – rather than excess mortality among surveyed cases, which was also noted in a different district in Niger in 2017 [7].

The investigators used very basic tools to assess the presence of after-effects, or relied on respondents' statements. In the case of hearing loss, for example, it might have been preferable to use a formal audiometric assessment to identify more subtle hearing problems. However, these tools are not readily available in Niger. Future studies in the area should also focus on the experiences of families and caregivers of persons with sequelae, the economic impact of these sequelae, and educational and occupational impacts on patients. Worse still, even for the sequelae identified in this survey, recourse to rehabilitation care or medical specialists is limited. One of the underlying objectives of this survey was to provide a better description of the current burden of sequelae so that more appropriate epidemic responses could be organised in the future. It would therefore seem important to focus efforts on audiometry and hearing aids, improved mobility and housing for people with paralysis, and a programme to manage epilepsy, among others. Given the epidemiological profile of cases, specific programming should also be planned for the paediatric population.

## Conclusion

The Zinder region experienced an epidemic of meningococcal meningitis in 2022. This epidemic followed several years of epidemics in neighbouring areas. We found a high prevalence of debilitating sequelae with little access to rehabilitation services. While we hope that such epidemics will not occur again in the years to come, responses to epidemics of meningococcal meningitis must be adapted to identify sequelae of meningitis as early as possible and to orient people with disabilities to appropriate aftercare.

## Supporting information

**S1 Fig. Epidemic curves in the 5 Health Areas that crossed the epidemic threshold during the epidemic.** NB: The red lines represent the epidemic threshold in a HA; the orange lines represent the alert threshold in a HA.
(DOCX)

## Acknowledgments

The authors thank the participants and their families for their participation in this survey in sometimes challenging situations, and Nora Groce for her thoughtful comments on an early draft of this manuscript.

## Author contributions

**Conceptualization:** Georges Tonamou, Iza Ciglenecki, Matthew E Coldiron.

**Formal analysis:** Matthew E Coldiron.

**Investigation:** Abdoul-Aziz Idrissa, Salifou Atti, Roger Kiamvu Wasaulua, Ousmane Sani, Matthew E Coldiron.

**Methodology:** Matthew E Coldiron.

**Project administration:** Abdoul-Aziz Idrissa, Iza Ciglenecki, Matthew E Coldiron.

**Supervision:** Abdoul-Aziz Idrissa, Salifou Atti, Roger Kiamvu Wasaulua, Serge Kazadi, Elhadji Ibrahim Tassiou, Ousmane Guindo, Georges Tonamou, Iza Ciglenecki, Matthew E Coldiron.

**Validation:** Matthew E Coldiron.

**Writing – original draft:** Matthew E Coldiron.

**Writing – review & editing:** Abdoul-Aziz Idrissa, Salifou Atti, Roger Kiamvu Wasaulua, Serge Kazadi, Ousmane Sani, Elhadji Ibrahim Tassiou, Ousmane Guindo, Georges Tonamou, Iza Ciglenecki, Matthew E Coldiron.

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
