## [Decision Letter · Decision Letter 0]

5 Nov 2024

PONE-D-24-43633Sequelae following an epidemic of meningococcal meningitis in Niger in 2022PLOS ONE

Dear Dr. Coldiron,

Thank you for submitting your manuscript to PLOS ONE. After careful consideration, we feel that it has merit but does not fully meet PLOS ONE’s publication criteria as it currently stands. Therefore, we invite you to submit a revised version of the manuscript that addresses the points raised during the review process. Both reviewers identified parts of your manuscript that need clarification to better understand your data. 

We look forward to receiving your revised manuscript.

Kind regards,

Kelli L. Barr, Ph.D.

Academic Editor

PLOS ONE

 Journal requirements: When submitting your revision, we need you to address these additional requirements. 1. Please ensure that your manuscript meets PLOS ONE's style requirements, including those for file naming. The PLOS ONE style templates can be found at https://journals.plos.org/plosone/s/file?id=wjVg/PLOSOne_formatting_sample_main_body.pdf and https://journals.plos.org/plosone/s/file?id=ba62/PLOSOne_formatting_sample_title_authors_affiliations.pdf 2. Please include a complete copy of PLOS’ questionnaire on inclusivity in global research in your revised manuscript. Our policy for research in this area aims to improve transparency in the reporting of research performed outside of researchers’ own country or community. The policy applies to researchers who have travelled to a different country to conduct research, research with Indigenous populations or their lands, and research on cultural artefacts. The questionnaire can also be requested at the journal’s discretion for any other submissions, even if these conditions are not met.  Please find more information on the policy and a link to download a blank copy of the questionnaire here: https://journals.plos.org/plosone/s/best-practices-in-research-reporting. Please upload a completed version of your questionnaire as Supporting Information when you resubmit your manuscript.” Athira J K(Newgen) 8 Oct 2024: **Straive, at PRTC please request the following from the authors and ping me with follow up:“In the ethics statement in the Methods, you have specified that verbal consent was obtained. Please provide additional details regarding how this consent was documented and witnessed, and state whether this was approved by the IR. 3. We note that the grant information you provided in the ‘Funding Information’ and ‘Financial Disclosure’ sections do not match.  When you resubmit, please ensure that you provide the correct grant numbers for the awards you received for your study in the ‘Funding Information’ section.

Reviewers' comments:

Reviewer's Responses to Questions

**Comments to the Author**

1. Is the manuscript technically sound, and do the data support the conclusions?

Reviewer #1: Yes

Reviewer #2: Yes

2. Has the statistical analysis been performed appropriately and rigorously? 

Reviewer #1: Yes

Reviewer #2: Yes

3. Have the authors made all data underlying the findings in their manuscript fully available?

Reviewer #1: Yes

Reviewer #2: No

4. Is the manuscript presented in an intelligible fashion and written in standard English?

Reviewer #1: Yes

Reviewer #2: Yes

5. Review Comments to the Author

Reviewer #1: This manuscript is a very well written account of sequelae following an epidemic of meningococcal disease in Niger in 2022 due to serogroup C. It is very timely as there is short fall in data on sequelae in developing countries. This will be of benefit in the WHO roadmap of defeating meningitis by 2030.

Minor Comments

1. Was this all presenting as meningitis or meningitis/septicaemia?

2. Add the manufacturer of the ACW polysaccharide vaccine & add narrative as to why a polysaccharide vaccine for MenC was used in under 2 years of age when it is not efficacious?

3. Add a comment on the delay with administration of polysaccharide vaccination ie over 14 weeks and the benefits of conjugate vaccine eg MenFive that is now being utilised in Nigeria.

4. Where any strains cultured ie is the clonal complex of the the MenC strain known, if so please add.

Reviewer #2: Review of Sequelae following an epidemic of meningococcal meningitis in Niger in 2022

The manuscript ‘Sequelae following an epidemic of meningococcal meningitis in Niger in 2022’ describes a follow-up survey of survivors of a meningococcal serogroup C epidemic in Niger to determine the burden of sequelae in this setting. The paper is very well written and clearly describes in detail the methodology used as well as the findings and implications. These results provide a valuable insight into the lasting impact of meningococcal epidemics in low income countries. It would be of interest to epidemiologists, public health practitioners and health policy experts. I have a few suggestions that need to be addressed before the paper is accepted.

Major comments

Results: The results state there were 61/521 living patients with sequelae (11.7%), however the total number of cases in Table 5 is 67 (29+15+9+6+4+4) so the sequelae rate would be 67/521=12.9%. Please correct or explain the difference in the figures.

Results: The headline sequelae rate for all of the 521 living cases was 12-13% but we know this includes cases caused by other bacteria (and maybe viruses). It would be interesting to see what the sequelae burden/profile was for the 149 confirmed MenC cases only. Table 4 states that 18% of MenC confirmed cases has sequelae (which is higher than the overall 12-13% rate). Please could you add in an additional column to Table 5 to describe the sequelae profile of just the MenC cases. Although the case numbers are lower, it will give us a strain-specific view of the impact of the meningococcal epidemic, without the extra noise associated with the other non-MenC cases.

Line 228-232: The authors state that 27 additional cases that weren’t on the initial linelisting were identified during the follow-up survey. But these don’t seem to be included in the analysis itself (the 570 number is used, not the 597). Please make it clear and explain if and why they weren’t included.

Lines 244-249: Do you have any information on timing of vaccination? The results states how many cases were vaccinated but it doesn’t state whether the vaccines were given before or after disease onset. Figure 1 suggests there were cases after the vaccination campaign began, so could there have been vaccine failures/break through cases (i.e. cases in vaccinated individuals)?

Lines 244-249: The authors included vaccination data only for the 165 confirmed cases, but are there data available for the non-confirmed followed-up cases (i.e. the 570)? If so, please add these in.

Table 4: I suggest another row is added in under the PCR results with the data for the cases that weren’t tested/laboratory confirmed (maybe change “PCR results” to “Laboratory confirmation”). This will better illustrate how the confirmed cases compare to those that weren’t laboratory confirmed in terms of % with sequelae.

Line 260: “Prophylaxis was reported by at least one person in 215 of the 570 households (38%)”. But the 570 figure is for individual cases, not households. So the percentage figure is incorrect. What was the total number of households included in the analysis? This needs to be stated and the percentage figure corrected.

Minor comments

Line 54-55. “in Africa” repeated, would rephrase

Line 57: would add ‘typically’- “typically presenting with sudden onset fever with headache and stiff neck”

Line 61- need reference(s) for the 5-10% figure.

6. PLOS authors have the option to publish the peer review history of their article (what does this mean? ). If published, this will include your full peer review and any attached files.

**Do you want your identity to be public for this peer review?** For information about this choice, including consent withdrawal, please see our Privacy Policy .

Reviewer #1: **Yes: ** Ray Borrow

Reviewer #2: No

---

## [Author Response · Author response to Decision Letter 1]

20 Dec 2024

Reviewer #1: This manuscript is a very well written account of sequelae following an epidemic of meningococcal disease in Niger in 2022 due to serogroup C. It is very timely as there is short fall in data on sequelae in developing countries. This will be of benefit in the WHO roadmap of defeating meningitis by 2030.

Authors’ response:

Thank you for this comment.

Minor Comments

1. Was this all presenting as meningitis or meningitis/septicaemia?

Authors’ response:

During the epidemic, the only case definitions used in health facilities were those referred to in the article, the usual WHO case definition. It is therefore possible that cases with meningococcaemia alone were therefore not captured in the surveillance system.

2. Add the manufacturer of the ACW polysaccharide vaccine & add narrative as to why a polysaccharide vaccine for MenC was used in under 2 years of age when it is not efficacious?

Authors’ response:

We have added the manufacturer (Finlay) and corrected the error – the campaign indeed targeted persons 2-29 years of age.

3. Add a comment on the delay with administration of polysaccharide vaccination ie over 14 weeks and the benefits of conjugate vaccine eg MenFive that is now being utilised in Nigeria.

Authors’ response:

We thank the reviewer for this comment – and have added this point into the discussion.

4. Where any strains cultured ie is the clonal complex of the the MenC strain known, if so please add.

Authors’ response:

Unfortunately, these data were not available. Where culture was performed, results were not linked into the databases that were used for this study.

Reviewer #2: Review of Sequelae following an epidemic of meningococcal meningitis in Niger in 2022

The manuscript ‘Sequelae following an epidemic of meningococcal meningitis in Niger in 2022’ describes a follow-up survey of survivors of a meningococcal serogroup C epidemic in Niger to determine the burden of sequelae in this setting. The paper is very well written and clearly describes in detail the methodology used as well as the findings and implications. These results provide a valuable insight into the lasting impact of meningococcal epidemics in low income countries. It would be of interest to epidemiologists, public health practitioners and health policy experts. I have a few suggestions that need to be addressed before the paper is accepted.

Major comments

Results: The results state there were 61/521 living patients with sequelae (11.7%), however the total number of cases in Table 5 is 67 (29+15+9+6+4+4) so the sequelae rate would be 67/521=12.9%. Please correct or explain the difference in the figures.

Authors’ response:

We thank the reviewer for catching this. Both the proportion reported in the text and the proportions in the table are correct: some patients reported more than one sequela. This was obliquely referenced in the text, and we have added clarification in the Table legend to ensure that this is more easily understood.

Results: The headline sequelae rate for all of the 521 living cases was 12-13% but we know this includes cases caused by other bacteria (and maybe viruses). It would be interesting to see what the sequelae burden/profile was for the 149 confirmed MenC cases only. Table 4 states that 18% of MenC confirmed cases has sequelae (which is higher than the overall 12-13% rate). Please could you add in an additional column to Table 5 to describe the sequelae profile of just the MenC cases. Although the case numbers are lower, it will give us a strain-specific view of the impact of the meningococcal epidemic, without the extra noise associated with the other non-MenC cases.

Authors’ response:

We thank the reviewer for this comment. We have added two columns (n and %) to Table 5, and added a sentence in the discussion section, as indeed, the prevalence of deafness was 10% and paralysis 6.4% among those with confirmed NmC, which is a point worth highlighting.

Line 228-232: The authors state that 27 additional cases that weren’t on the initial linelisting were identified during the follow-up survey. But these don’t seem to be included in the analysis itself (the 570 number is used, not the 597). Please make it clear and explain if and why they weren’t included.

Authors’ response:

These cases were not included in our clinical descriptions as there was no additional information available regarding their clinical course (because they were not in the linelist). We have added a sentence in the results section to clarify this and justify their exclusion from the descriptions.

Lines 244-249: Do you have any information on timing of vaccination? The results states how many cases were vaccinated but it doesn’t state whether the vaccines were given before or after disease onset. Figure 1 suggests there were cases after the vaccination campaign began, so could there have been vaccine failures/break through cases (i.e. cases in vaccinated individuals)?

Authors’ response:

Thanks for this point – which has been a recurring question in this area which has received multiple vaccination campaigns. Specific dates of vaccination were difficult to obtain in the survey – few participants had vaccination cards. In the four cases that did have dates, all vaccinations happened post-illness in 2022. In the end, it’s a very interesting question but not one that we’re really able to comment on given the data at our disposal.

Lines 244-249: The authors included vaccination data only for the 165 confirmed cases, but are there data available for the non-confirmed followed-up cases (i.e. the 570)? If so, please add these in.

Authors’ response:

We have added this information at the reviewer’s suggestion. There were no differences in vaccination coverage between confirmed and non-confirmed cases.

Table 4: I suggest another row is added in under the PCR results with the data for the cases that weren’t tested/laboratory confirmed (maybe change “PCR results” to “Laboratory confirmation”). This will better illustrate how the confirmed cases compare to those that weren’t laboratory confirmed in terms of % with sequelae.

Authors’ response:

These data are already presented as the last line of Table 4 in the row “PCR not performed”. We have changed the title of that section of the table to “Laboratory confirmation” at the reviewer’s suggestion and hope this will help clarify.

Line 260: “Prophylaxis was reported by at least one person in 215 of the 570 households (38%)”. But the 570 figure is for individual cases, not households. So the percentage figure is incorrect. What was the total number of households included in the analysis? This needs to be stated and the percentage figure corrected.

Authors’ response:

Thanks to the reviewer for pointing out this error, which has now been corrected. 534 individual households were represented.

Minor comments

Line 54-55. “in Africa” repeated, would rephrase

Authors’ response:

The second “in Africa” has been deleted.

Line 57: would add ‘typically’- “typically presenting with sudden onset fever with headache and stiff neck”

Authors’ response:

We have updated.

Line 61- need reference(s) for the 5-10% figure.

Authors’ response:

The reference has been added.

---

## [Decision Letter · Decision Letter 1]

14 Mar 2025

PONE-D-24-43633R1Sequelae following an epidemic of meningococcal meningitis in Niger in 2022PLOS ONE

Dear Dr. Coldiron,

Thank you for submitting your manuscript to PLOS ONE. After careful consideration, we feel that it has merit but does not fully meet PLOS ONE’s publication criteria as it currently stands. Therefore, we invite you to submit a revised version of the manuscript that addresses the minor points raised during the review process.

We look forward to receiving your revised manuscript.

Kind regards,

Ray Borrow, Ph.D., FRCPath

Academic Editor

PLOS ONE

Journal Requirements:

Reviewers' comments:

Reviewer's Responses to Questions

**Comments to the Author**

1. If the authors have adequately addressed your comments raised in a previous round of review and you feel that this manuscript is now acceptable for publication, you may indicate that here to bypass the “Comments to the Author” section, enter your conflict of interest statement in the “Confidential to Editor” section, and submit your "Accept" recommendation.

Reviewer #2: All comments have been addressed

Reviewer #3: (No Response)

2. Is the manuscript technically sound, and do the data support the conclusions?

Reviewer #2: Yes

Reviewer #3: Yes

3. Has the statistical analysis been performed appropriately and rigorously? 

Reviewer #2: Yes

Reviewer #3: Yes

4. Have the authors made all data underlying the findings in their manuscript fully available?

Reviewer #2: Yes

Reviewer #3: Yes

5. Is the manuscript presented in an intelligible fashion and written in standard English?

Reviewer #2: Yes

Reviewer #3: Yes

6. Review Comments to the Author

Reviewer #2: (No Response)

Reviewer #3: Please find attached Reviewer Comments. A couple of comments and suggestions have been highlighted to be effected by the authors.

7. PLOS authors have the option to publish the peer review history of their article (what does this mean? ). If published, this will include your full peer review and any attached files.

**Do you want your identity to be public for this peer review?** For information about this choice, including consent withdrawal, please see our Privacy Policy .

Reviewer #2: No

Reviewer #3: **Yes: ** George Gyamfi-Brobbey

---

## [Author Response · Author response to Decision Letter 2]

28 Mar 2025

Line 24 – 26: Please break the first sentence into 2 parts – seems too long and contain 2 main info.

Authors’ response: We thank the reviewer for these positive comments.

Line 71 – 75: are you able to state the current treatment guidelines in Niger for treating sequelae following meningococcal meningitis. If Corticosteroids are prescribed by some doctors, do they do so by their own discretion?

Authors’ response: We have added additional information that neither empiric steroid use nor standard sequelae screening are part of current Nigerien guidelines.

Line 80 – 82: is/was there a follow up to ascertain the effectiveness of the vaccination campaign in this cohort, especially in those under 5 years?

Authors’ response: We have specified that no formal VE evaluations took place following the polysaccharide campaigns of 2021 and 2022.

Lines 93 and 97: “Suspected case” rather?

Authors’ response: We have corrected this terminology.

Line 114: Consider “Verbally” informed consent……

Authors’ response: We have corrected this terminology.

Line 125 – 126: please change to 16th and 30th November 2022; and similar references throughout the script.

Authors’ response: These changes have been made throughout.

Line 172 – 174: Can you please add the complete in-text reference, year of publication of these ethical guidelines and owners/authors.

Authors’ response: This reference has been added.

Line 193: 1001, comma not needed in the figure.

Authors’ response: This has been deleted.

Line 297: Please delete “during the epidemic.”

Authors’ response: This has been deleted.

---

## [Editor Report · Decision Letter 2]

6 Apr 2025

Sequelae following an epidemic of meningococcal meningitis in Niger in 2022

PONE-D-24-43633R2

Dear Dr. Coldiron,

We’re pleased to inform you that your manuscript has been judged scientifically suitable for publication and will be formally accepted for publication once it meets all outstanding technical requirements.

Kind regards,

Ray Borrow, Ph.D., FRCPath

Academic Editor

PLOS ONE
---

## [Editor Report · Acceptance letter]

PONE-D-24-43633R2

PLOS ONE

Dear Dr. Coldiron,

I'm pleased to inform you that your manuscript has been deemed suitable for publication in PLOS ONE. Congratulations! Your manuscript is now being handed over to our production team.

Kind regards,

on behalf of

Prof. Ray Borrow

Academic Editor

PLOS ONE